# Differences in the Inflammatory Response of White Adipose Tissue and Adipose-Derived Stem Cells

**DOI:** 10.3390/ijms21031086

**Published:** 2020-02-06

**Authors:** Sara Taha, Elias Volkmer, Elisabeth Haas, Paolo Alberton, Tobias Straub, Diana David-Rus, Attila Aszodi, Riccardo Giunta, Maximilian Michael Saller

**Affiliations:** 1Experimental Surgery and Regenerative Medicine (ExperiMed), Department of General, Trauma and Reconstructive Surgery, Ludwig-Maximilians-University (LMU), Fraunhoferstraße 20, 82152 Planegg-Martinsried, Germany; sara.taha@med.uni-muenchen.de (S.T.); elias.volkmer@helios-gesundheit.de (E.V.); elisabeth.haas@med.uni-muenchen.de (E.H.); paolo.alberton@med.uni-muenchen.de (P.A.); attila.aszodi@med.uni-muenchen.de (A.A.); 2Division of Hand, Plastic and Aesthetic Surgery, Ludwig-Maximilians-University (LMU), Pettenkoferstraße. 8a, 80336 Munich, Germany; 3Department of Hand Surgery, Helios Klinikum München West, Steinerweg 5, 81241 Munich, Germany; 4Bioinformatics Unit, Biomedical Center Munich, Ludwig-Maximilians-University (LMU), Großhaderner Straße 9, 82152 Planegg-Martinsried, Germany; tobias.straub@med.uni-muenchen.de; 5Institute for Medical Information Processing, Biometry, and Epidemiology (IBE), Ludwig-Maximilians-University (LMU), Marchioninistr. 15, 81377 Munich, Germany; ddavidrus@ibe.med.uni-muenchen.de

**Keywords:** white fat tissue, adipose-derived stem cells, immunomodulation, inflammation, TNFalpha

## Abstract

The application of liposuctioned white adipose tissue (L-WAT) and adipose-derived stem cells (ADSCs) as a novel immunomodulatory treatment option is the currently subject of various clinical trials. Because it is crucial to understand the underlying therapeutic mechanisms, the latest studies focused on the immunomodulatory functions of L-WAT or ADSCs. However, studies that examine the specific transcriptional adaptation of these treatment options to an extrinsic inflammatory stimulus in an unbiased manner are scarce. The aim of this study was to compare the gene expression profile of L-WAT and ADSCs, when subjected to tumor necrosis factor alpha (TNFα), and to identify key factors that might be therapeutically relevant when using L-WAT or ADSCs as an immuno-modulator. Fat tissue was harvested by liposuction from five human donors. ADSCs were isolated from the same donors and shortly subjected to expansion culture. L-WAT and ADSCs were treated with human recombinant TNFα, to trigger a strong inflammatory response. Subsequently, an mRNA deep next-generation sequencing was performed to evaluate the different inflammatory responses of L-WAT and ADSCs. We found significant gene expression changes in both experimental groups after TNFα incubation. However, ADSCs showed a more homogenous gene expression profile by predominantly expressing genes involved in immunomodulatory processes such as *CCL19*, *CCL5*, *TNFSF15* and *IL1b* when compared to L-WAT, which reacted rather heterogeneously. As RNA sequencing between L-WAT and ADSCS treated with TNFα revealed that L-WAT responded very heterogeneously to TNFα treatment, we therefore conclude that ADSCs are more reliable and predictable when used therapeutically. Our study furthermore yields insight into potential biological processes regarding immune system response, inflammatory response, and cell activation. Our results can help to better understand the different immunomodulatory effects of L-WAT and ADSCs.

## 1. Introduction

Inflammation is a complex, multifaceted state for many chronic conditions. The ability to regulate an adequate inflammatory response is pivotal to prevent the development and progression of any disease. Inflammatory processes are characterized by an interplay between pro- and anti-inflammatory cytokines. Cytokines, such as interleukin-1 (IL-1), tumor necrosis factor (TNF) and gamma-interferon (IFN-γ), are classified as pro-inflammatory, whereas IL-4, IL-10 and IL-14 are classified as anti-inflammatory cytokines [1,2,3]. However, this classification is far too simplistic, since most cytokines may act as a pro- as well as an anti-inflammatory cytokine [1,4]. In many chronic diseases, such as osteoarthritis, rheumatoid arthritis and Crohn’s disease, the balance of pro- and anti-inflammatory cytokines shifts toward pro-inflammatory factors and ultimately requires treatment [5]. Thus, novel treatment options targeting cytokine imbalance in inflammatory conditions are under investigation.

Human white fat cells and stem cells, derived of white fat tissue (ADSCs), have been shown to exert immunomodulatory effects both in vitro and in vivo [6,7,8]. White adipose tissue (WAT) is a highly complex organ. Rather than functioning as a mere energy storage, it also plays a potent role in metabolic and endocrine balance. It is composed of adipocytes, loose connective tissue matrix and the stromal vascular fraction (SVF). The SVF consists of preadipocytes, capillary endothelial cells, infiltrated monocytes/macrophages and a small subpopulation of multipotent ADSCs. The cellular composition, cell size and cell activity are highly variable and dependent on the donor and the tissue source [9,10,11]. Similar therapeutic effects are described for both WAT and ADSCs, yet there is a trend in regenerative and immunomodulatory medicine toward the utilization of the supposedly more versatile ADSCs. They are thought to be the most promising cells of the SVF regarding medical benefits [12,13,14,15,16,17], and their molecular features make them promising candidate cells, not only in the field of regenerative medicine, but also for the treatment of inflammatory-related disorders [7,18,19,20,21,22,23,24]. ADSCs can exert immunomodulation through direct contact with immune cells or by secretion of paracrine factors [5,7,8]. Dependent on their microenvironment, ADSCs exhibit pro-inflammatory and anti-inflammatory properties [6]. ADSCs have the ability to interact with many components of the innate immune system, including soluble complement, macrophages, dendritic cells, neutrophils, mast cells and natural killer cells [6,25,26]. Furthermore, ADSCs have the capacity to interfere with the adaptive immune system [27,28,29]. For clinical purposes, ADSCs have been used as anti-inflammatory “bioreactors” in the case of inflammatory bowel diseases, osteoarthritis, diabetes mellitus, chronic wounds and scar treatment [13,14,15,19,30,31,32]. Nevertheless, there is still some ambiguity regarding the utilization of ADSCs as a cellular treatment option. Potentially harmful short- or long-term effects may still be discovered, and as the exact working mechanisms are yet to be unraveled, the application as a treatment option is legally restricted in most countries.

In contrast, the use of simple WAT for clinical purposes is not subject to such strict legal restrictions in most countries, as long as it is used as an unpurified autologous lipo-transfer. Current clinical applications include scar treatment, breast reconstruction after breast cancer surgery, aesthetic rejuvenation, cleft-lip repair and liposculpture for body deformities [30,33,34,35,36]. The treatment of bone defects, osteomyelitis or chronic wounds is being debated [13,30,35,36,37,38,39]. Although having a higher immunomodulatory and regenerative potential, the clinical use of purified ADSCs is, as mentioned above, even more restricted. Given the huge potential of treating chronic inflammatory processes with the more appealing ADSCs, the mechanisms of action need to be further explored in order to facilitate legalization of future clinical applications.

For this purpose, we aimed to generate an unbiased dataset of transcriptional changes after an extrinsic inflammatory stimulus. Therefore, liposuctioned white fat tissue (L-WAT) and ADSCs from the same donor were treated with tumor necrosis factor alpha (TNFα) in vitro. Subsequently, a bioinformatic analysis of obtained RNA sequencing was performed to evaluate the different inflammatory responses of L-WAT and ADSCs.

## 2. Results

### 2.1. Gene Expression after TNFα Treatment of ADSCs Is More Homogenous in Comparison to L-WAT

To assess the immunomodulatory response of L-WAT and ADSPC of the same donor after TNFα exposure, we performed a deep RNA-sequencing (Figure 1).

TNFα induced significant changes in gene expression in L-WAT and ADSCs, when compared to controls. After normalization and correction for multiple testing, differential gene expression analysis revealed 83 genes that were significantly upregulated and 29 genes that were significantly downregulated in L-WAT treated with TNFα, when compared to untreated L-WAT. Interestingly, ADSCs from different donors showed a considerably more homogenous transcriptional response, which results in 1404 and 1109 genes that were significantly upregulated and downregulated, respectively, in ADSCs treated with TNFα, when compared to ADSCs in normal culture conditions. Common to both groups, 68 genes were significantly upregulated, and 11 genes were significantly downregulated. While Table 1 shows the five most TNF-dependent upregulated and downregulated genes, when comparing ADSCs to ADSCs, and TNFα or L-WAT to L-WAT and TNFα (Table 1), Appendix A includes all significantly changed genes of ADSCs and L-WAT after TNFα treatment.

### 2.2. Gene Expression Regulation in Fat and ADSCs Treated with TNFα

To further highlight the different gene expression response of L-WAT and ADSCs to TNFα treatment, we analyzed the significantly upregulated and downregulated genes in the L-WAT vs. ADSCs, as well as in the experimental group L-WAT and TNFα vs. ADSCs and TNFα. We found 3388 genes significantly higher expressed in ADSCs, in contrast to L-WAT. Furthermore, our evaluation showed 2397 significantly higher expressed genes in ADSCs and TNFα, when compared to L-WAT and TNFα. Interestingly, 699 genes out of the 2397 significantly upregulated genes in ADSCs were TNFα-dependent (Figure 2A). In comparison, 3129 genes were significantly lower expressed in ADSCs, when compared to L-WAT. After incubation with TNFα, 3047 genes were significantly downregulated in ADSCs, when compared to L-WAT. Hereof, 1160 genes were significantly downregulated in ADSCs and TNFα, when compared to L-WAT and TNFα, due to the effect of TNFα. The remaining 1887 significantly downregulated genes in ADSCs and TNFα, when compared to L-WAT and TNFα were not related to the effect of TNFα (Figure 2B). The whole list of significantly changed genes in ADSCs and TNFα vs. L-WAT and TNFα is provided in Appendix A.

While a hierarchical cluster analysis of the principal component 1 (PC1), revealed a clear separation of ADSCs and L-WAT, PC4 clearly shows that ADSCs have a substantial higher homogenous gene expression profile, when compared to L-WAT (Figure 3). Moreover, ADSCs showed greater transcriptome changes after incubation with TNFα, when compared to L-WAT (Figure 3, PC4).

The 30 most to PC1 contributing genes are fat-related marker genes, including *LEP*, *FABP4* and *ADIPOQ*. Furthermore, it revealed six genes that are higher expressed in ADSCs; these include *GREM1*, known to be involved in limb development [40], *CNN1*, which plays a role in smooth-muscle function [41], and *ALPK2*, which is important for cardiac muscle cell development [42] (Figure 4A).

Interestingly, the analysis of the fourth cluster (PC4) revealed the separation among genes associated with immunomodulatory processes such as *CX3CL1*, *IL-4I1*, *IL-31* and *CCL5*. (Figure 4B). However, this separation was strongly visible in ADSCs, whereas, in L-WAT, the inflammatory effect was not as strong. The top five prominent genes in this separation are as follows: BIRC3, which is important for the inhibition of apoptosis [43]; *MEOX1*, which plays a role in sclerotome development [44]; *CX3CL1*, which is pivotal for chemotaxis and cell adhesion [45]; *CCL19*, which plays a crucial role in different inflammatory processes [46,47]; and *ANO9*, which might play a role in different types of cancer [48].

### 2.3. Biological Pathways that Are Regulated Upon TNFα Exposure in L-WAT and ADSCs

As TNFα is a pleiotropic cytokine with important functions, such as homeostasis, inflammation, pathogenesis, apoptosis or necroptosis [49], different biological processes were significantly changed in the experimental groups after exposure to TNFα. We carried out pathway and functional analysis, using Gene Ontology (GO), including all genes that were differentially regulated upon L-WAT and ADSCS treated with TNFα (ADSCs and TNFα vs. L-WAT and TNFα). The evaluation with GO of all significantly higher regulated genes in ADSCs, when compared to L-WAT treated with TNFα, revealed a plentitude of different biological functions, like immune system processes, extracellular matrix organization and response to an inflammatory stimulus (Figure 5A). Furthermore, the analysis with GO of significantly lower-expressed genes in ADSCs, when compared to L-WAT treated with TNFα, showed alterations in different biological pathways, including developmental processes, biological adhesion and leukocyte migration (Figure 5B). The whole list of significantly different biological processes is provided in Appendix A.

To get a better understanding of the differentially regulated pathways in ADSCs and L-WAT upon exposure to TNFα, we performed a gene set enrichment analysis (GSEA). Interestingly, ADSCs and TNFα showed a significant increase of inflammation-related gene hallmarks, as well as gene hallmarks, like epithelial to mesenchymal transition and apical junctions, when compared to L-WAT and TNFα (Figure 6). Intriguingly, while TNFα exposure of ADSCs led to interferon alpha-, as well as gamma-related response, L-WAT showed a mild response in the interferon gamma pathway and nearly no gene-set enrichment in the interferon alpha hallmark (Figure 6, blue arrowheads). Additionally, the most significantly underrepresented gene sets of ADSCs and TNFα mainly consist of fat-related metabolism and development sets, when compared to L-WAT and TNFα (Figure 6).

## 3. Discussion

L-WAT and ADSCs are seen as a promising therapy tools in the field of regenerative medicine. While the use of simple fat tissue is straightforward, the therapeutic use of isolated stem cells is controversial and, to date, not clinically approved in most countries. Because of the inflammatory component, which characterize several clinical conditions, it is of upmost importance to discover novel immunomodulatory treatment options and to understand their mechanisms of action. To date, there are several studies that investigated the immunomodulatory function of (L-)WAT and ADSCs, but to our knowledge, there is none specifically investigating the differences in both in an unbiased manner [20,50,51,52]. In the early 2000s, Zuk et al. described the in vitro potential of human ADSCs to differentiate under specific culture conditions into different mesenchymal cell linages [16,53]. Later, mesenchymal stem cells and ADSCs were introduced as trophic mediators for tissue repair, and it was proposed that they secrete factors that stimulate the release of functional bioactive factors from surrounding cells [54,55]. This view has been evolved, and MSCs and ADSCs are now believed to secrete paracrine factors themselves that promote cell viability, proliferation and matrix production in the surrounding environment [55]. Different studies show that the secretome of ADSCs, exerted through extracellular vesicles, is a promising source of new cell-free therapies in the field of regenerative medicine [24,56,57,58,59,60,61]. The identification of the exact overall immunomodulatory response of L-WAT and ADSCs is crucial for clinical approaches, in order to introduce targeting therapies. Therefore, our aim was to investigate the differences in the inflammatory response of L-WAT and ADSCs.

In our study, we identified genes in L-WAT and ADSCs, as well as pathways induced or repressed in inflammation that are modulated by TNFα exposure and may represent candidates for targeting treatment in inflammatory conditions. As expected, the analysis detected genes specifically involved in TNFα-induced inflammatory processes. After correction for multiple testing, 5444 genes showed significant differential gene expression in ADSCs treated with TNFα, when compared to L-WAT treated with TNFα. Our data provide evidence that ADSCs display greater transcriptional changes after TNFα treatment, when compared to L-WAT (Figure 3). The PCA clearly shows that ADSCs display a more homogenous gene expression between cells isolated from different individuals, when compared to a strong heterogeneous gene expression profile in L-WAT. This fact might indicate that L-WAT is much more susceptible to inter-individual factors and thus might influence its therapeutic effect. This appears to be even more interesting, when considering the circumstance that isolated ADSCs and fat tissue were harvested from the same donors.

WAT is a whole tissue with its intrinsic and complex cellular and biochemical components, which makes its clinical use difficult. In addition, inter-individual donor factors, such as age, sex, ancestry and medical conditions, have a hardly predictable and yet unknown influence on the therapeutical outcome. Furthermore, WAT contains not only progenitor cells, but also adipocytes, blood cells, immune cells and soluble factors that can influence the gene expression profile. On the other hand, since WAT is composed of different components, it might be concluded that when used for therapeutic purposes, these components can synergistically exert their positive effects. Different studies showed that ADSCs cannot exert their claimed therapeutic effects solitary but need different “co-factors” [50]. Furthermore, ADSCs injected into osteoarthritic joints are not detectable anymore after a few days [62,63]. Therefore, it is assumed that ASPCs “imprint” their anti-inflammatory effects on cells of the immune system, which then give a prolonged ameliorating effect [15,62,63]. These findings indicate that a composition of progenitor cells and different cells of the immune system, as found in L-WAT, might have a stronger positive therapeutic effect than its individual injected factors. On the other hand, isolated ADSCs are referred to be the most promising and potent component of L-WAT, because these cells behave in a more predictable manner [13,17,20,64]. Different studies have shown that an inflammatory environment, as found in many chronic diseases, extensively enhances the immunosuppressive effects of ADSCs [65]. However, it remains uncertain if there is a special “threshold” that needs to be reached to activate the immunomodulatory effects of L-WAT and ADSCs. Therefore, searching for the strategies that can activate the trophic functions of L-WAT and ADSCs is fundamental for their application in regenerative medicine.

While an inflammatory response to TNFα treatment was shown in all donors, the effect of TNFα seemed to be stronger in ADSCs. This might be mostly due to the highly heterogeneous gene expression profile and the high proportion of fat-related gene sets in L-WAT treated with TNFα, which may alleviate its inflammatory effect. Another limitation of the presented study is the utilization of ADSCs and L-WAT from different anatomical regions and the in vitro expansion of ADSCs, as the complexity of the immunomodulatory action of ADSCs and WAT cannot be resolved by pure in vitro experiments [66]. Nonetheless, our GSE analysis revealed that gene sets for interferon alpha and gamma were differently regulated between ADSCs and L-WAT upon TNFα treatment. This novel result can be utilized in future experiments, by confining genes that are involved in the immunomodulatory properties of ADSCs. Future studies should focus on the direct transcriptional analysis of WAT in different inflammatory conditions (IFNα/β, IFNγ and/or TNFα), on a single-cell level, to obtain biologically relevant data. This experimental approach will help to unravel the interactions between the various cell types of WAT, including ADSCs.

## 4. Materials and Methods 

### 4.1. Ethics Statement and Sample Acquisition

After obtaining written informed consent, human L-WAT was obtained from five patients without systemic diseases (mean age: 47.4 years), undergoing water-jet-assisted liposuction with the Body-Jet system (human med AG, Germany) from subcutaneous regions, for aesthetic reasons. Liposuctions with the Body-Jet system were performed with 3.5 and/or 3.8 mm cannulas and a pressure of approximately 550 bar. This study was conducted in accordance with the declaration of Helsinki and approved by the ethics committee of Ludwig-Maximilians-University, Munich (275-16). All lipoaspirates were harvested from the abdomen or thighs, through liposuction by a surgeon following common surgical standards. All patients were previously screened and tested negative for HIV (human immunodeficiency virus), HCV (hepatitis C virus) and HBV (hepatitis B virus). Patients’ information is summarized in Appendix A.

### 4.2. Preparation of White Adipose Tissue, Cell Isolation and Culture Conditions

For all samples, a portion of 1.5 g L-WAT was washed twice with phosphate-buffered saline (PBS), to remove residual blood, and afterward they were directly incubated in standard culture medium, consisting of DMEM-high glucose (Thermo Fisher Scientific, USA) supplemented with 10% fetal bovine serum (FBS, Sigma-Aldrich, USA), 100 U/mL of Penicillin and 100 µg/mL of Streptomycin (Life Technology, USA), in a humidified incubator (21% O_2_, 5% CO_2_ and 37 °C), supplemented with 50 ng/mL of recombinant TNFα (Merck, Germany), for 48 h, to stimulate the secretion of inflammatory paracrine factors. ADSCs were isolated from approximately 10 g of the same L-WAT with a semi-automated centrifuge system (ARC™-Processing Unit, InGeneron, USA), following the manufacturer’s protocol and using its enzyme blend (Matrase™) and 37 °C warm lactated Ringer’s solution (Fresenius Kabi, Germany). Stem cell properties were proven by differentiating a proportion of isolated cells into the osteogenic, adipogenic and chondrogenic lineage, as previously published [67]. For RNA-Seq experiments, freshly isolated cells were expanded for 3 days in standard culture medium, as described above. After cell expansion for 72 h, 150,000 ADSCs in passage 1 were cultured for an additional 48 h in standard culture medium, supplemented with 50 ng/mL recombinant TNFα. ADSCs and L-WAT in standard culture medium in the absence of TNFα served as a control. ASCs after 72 h (ASC 0 h) and fresh L-WAT after washing (L-WAT 0 h) served as baseline controls. After cultivation, ADSCs and L-WAT were lysed in Trizol (Invitrogen, USA) and stored at −80 °C, until RNA sequencing.

### 4.3. Deep RNA-Sequencing and Bioinformatics

Total RNA was isolated by following a standardized protocol. RNA quality and quantity were measured with a BioAnalyzer (Agilent, USA), and libraries for sequencing were prepared with a SENSE mRNA-Seq Library Prep Kit V2 (Lexogen, Austria). All libraries were sequenced on a HiSeq1500 device (Illumina, USA) with a read length of 50 bp and a sequencing depth of approximately 20 million reads per sample. After demultiplexing, reads were aligned to the human reference genome (version GRCH38.85) with STAR (version 2.5.3a) [68]. Expression values (TPM) were calculated with RSEM (version 1.3.0) [69]. Genes detected in less than five samples were removed from further analysis. Differential gene expression analysis was performed, using the “voom” function in edgeR (version 3.26.5) [70] with a linear model encompassing biological condition and sequencing lane as fixed and random effect, respectively. An adjusted *p*-value (FDR) of less than 0.05 was set to determine significantly changed genes.

## 5. Conclusions

Autologous fat grafting has been carried out for decades as a standard procedure in many fields of plastic and reconstructive surgery, whereas the use of isolated ADSCs is highly restricted in most countries. In this study, we show that both L-WAT and ADSCs exhibit a strong response when exposed to an inflammatory environment. However, the inflammatory effect of TNFα on transcriptome regulation is more pronounced and predictable in ADSCs, when compared to L-WAT, which displays a very heterogeneous gene expression profile.

## Figures and Tables

**Figure 1 ijms-21-01086-f001:**
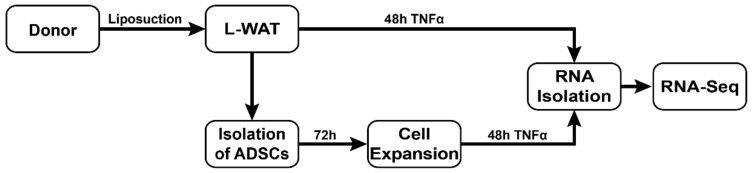
Experimental timeline and setup. Human L-WAT was harvested from five different donors through water-jet-assisted liposuction. Fresh fat samples were incubated with TNFα for 48 h, to mimic an inflammatory milieu that triggers a strong immune response. Simultaneously, ADSCs were isolated from the same donors’ fat samples harvested during earlier liposuction. After an expansion period of 72 h, ADSCs were also treated with TNFα. Subsequently the incubation with TNFα, RNA from fat and cell samples was isolated and sequenced. A portion of the same sample without TNFα was implemented in parallel and used as reference control.

**Figure 2 ijms-21-01086-f002:**
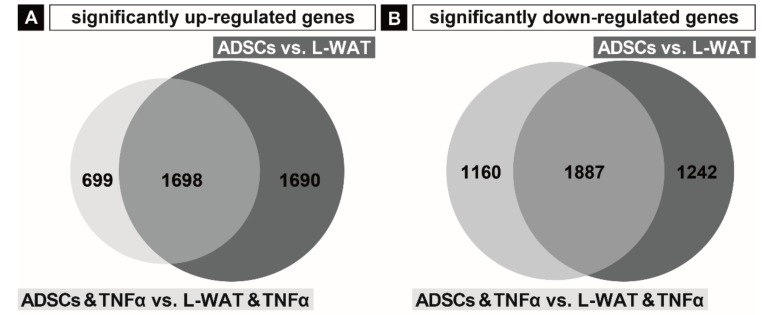
Significantly differentially expressed genes in L-WAT and ADSCs. Significantly upregulated genes in ADSCs compared to L-WAT and in ADSCs and TNFα compared to L-WAT and TNFα (**A**). Significantly, downregulated genes in ADSCs compared to L-WAT and in ADSCs and TNFα compared to L-WAT and TNFα (**B**).

**Figure 3 ijms-21-01086-f003:**
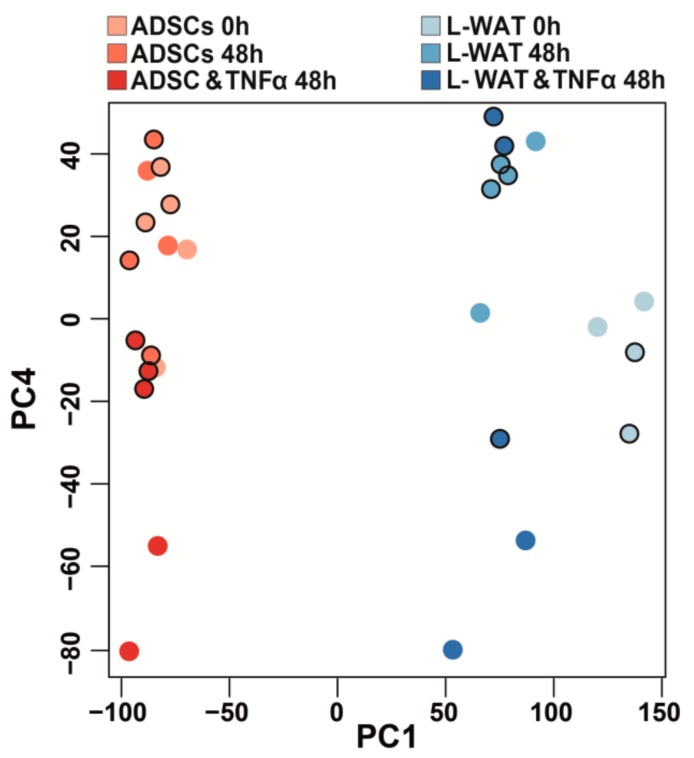
Principal Component Analysis (PCA) of L-WAT and ADSCs. The PCA revealed a clear separation of L-WAT (blue dots) and ADSCs (red dots) along the main component PC1. In addition, TNFα treatment showed a separation along PC4, with a more homogeneous response of ADSCs (dark-red dots), when compared to L-WAT (dark-blue dots). There was no clear clustering of samples from the abdomen (black border) or the thighs (no border).

**Figure 4 ijms-21-01086-f004:**
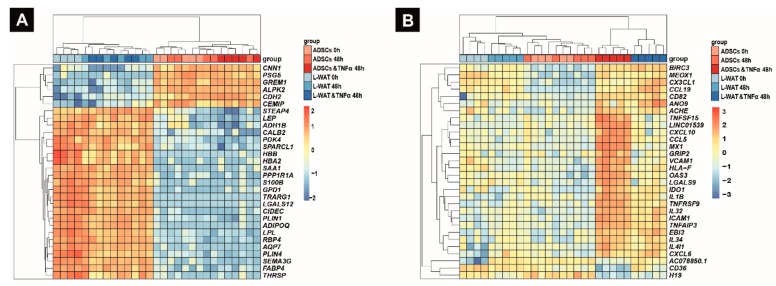
Hierarchical cluster analysis of the 30 most contributing genes of PC1 and PC4. The 30 most to PC1 contributing genes are fat-related marker genes including *LEP*, *FABP4* and *ADIPOQ* (**A**). The analysis of the PC4 reveals the separation among genes associated with immunomodulatory processes (**B**).

**Figure 5 ijms-21-01086-f005:**
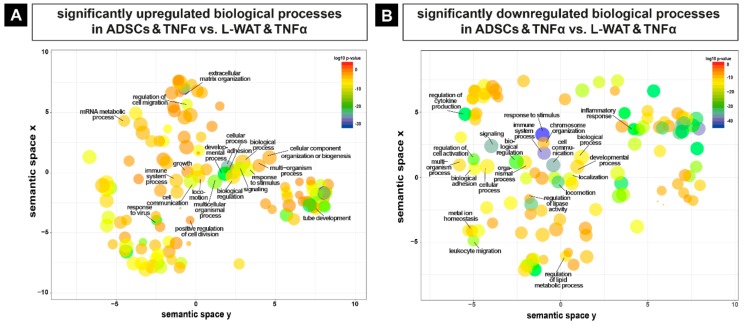
Significantly changed biological pathways in ADSCs and L-WAT after incubation with TNFα. Differential gene expression analysis revealed hundreds of significant upregulated (**A**) and downregulated (**B**) biological pathways in ADSCs treated with TNFα, when compared to L-WAT treated with TNFα.

**Figure 6 ijms-21-01086-f006:**
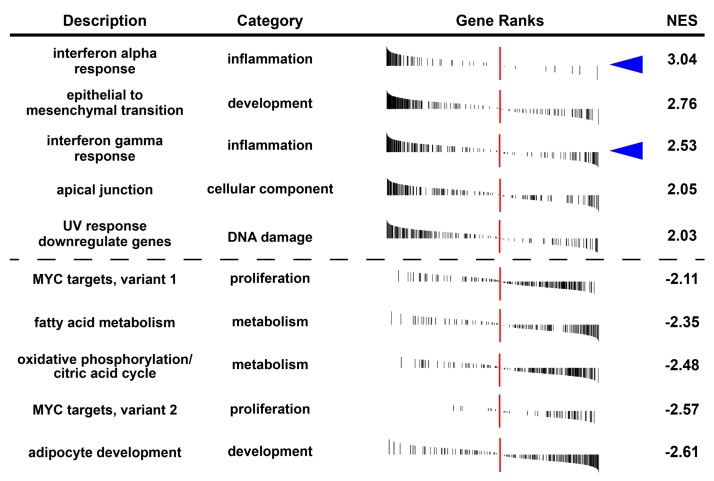
Visualization of the five most positively (NES > 0) or negatively (NES < 0) TNFα-dependent hallmark gene sets in ADSCs, when compared to L-WAT. Blue arrowheads indicate the difference of interferon alpha and gamma response of ADSCs and L-WAT upon TNFα treatment. NES: normalized enrichment score. Adjusted *p*-value < 0.05.

**Table 1 ijms-21-01086-t001:** Five most TNFα-dependent upregulated and downregulated in ADSCs and L-WAT; logFC: logarithmic fold-change.

ADSCs vs. ADSCs and TNFα		
**Gene symbol**	**Description**	logFC
*CXCL10*	C-X-C motif chemokine ligand 10	10.08
*CXCL11*	C-X-C motif chemokine ligand 11	9.67
*CCL5*	C-C motif chemokine ligand 5	9.60
*CXCL8*	C-X-C motif chemokine ligand 8	9.38
*LINC01539*	long intergenic non-protein coding RNA 1539	8.49
*PLA2G2A*	phospholipase A2 group IIA	−6.22
*WISP2*	WNT1 inducible signaling pathway protein 2	−6.47
*TNNT3*	troponin T3, fast skeletal type	−6.48
*ASPN*	Asporin	−7.06
*H19*	H19, imprinted maternally expressed transcript	−8.17
**L-WAT vs. L-WAT and TNFα**		
Gene symbol	Description	logFC
*CCL22*	C-C motif chemokine ligand 22	5.68
*ANO9*	anoctamin 9	5.30
*MMP9*	matrix metallopeptidase 9	5.17
*EBI3*	Epstein–Barr virus induced 3	4.94
*CCL5*	C-C motif chemokine ligand 5	4.68
*ECSCR*	endothelial cell surface expressed chemotaxis and apoptosis regulator	−2.97
*AC091939.1*	novel transcript	−3.13
*MNDA*	myeloid cell nuclear differentiation antigen	−3.57
*CA4*	carbonic anhydrase 4	−4.08
*AC002546.1*	novel transcript	−4.13

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
