# Peer review of "Differences in the Inflammatory Response of White Adipose Tissue and Adipose-Derived Stem Cells"

_ijms, 2020, doi:10.3390/ijms21031086_

Round 1

Reviewer 1 Report

The work of Taha et al. investigates differences in inflammatory response after stimulation with TNF-, between white adipose tissue (WAT) and adipose-derived stem / progenitor cells (ADSPCs). Although only paired samples from five donors were analyzed in this study, a very large number of genes were determined.

They found significant gene expression changes in both experimental groups after TNF- incubation. In addition, ADSPCs showed a more homogeneous gene expression profile by expressing predominantly genes involved in immunomodulatory processes compared with WAT. The study was well performed and written. Although it seems obvious that ADSPs are more functionally potent than WAT, it is relevant to assess this question for scientific reasons and in order to facilitate legalization of future clinical applications.

The authors should be some questions:

-The authors should be to clarify data on those five healthy donors, which are named as “healthy patients”, which seems a contradiction. It is important to clarify explicitly that these patients do not suffer from systemic diseases or if they a underwent liposuction by obesity. This information is important because they are factors that are known to influence the functional capacity of WAT and ADSPs. In addition, it is known that age is a factor that influences the heterogeneity of ADSPs. Therefore, the authors should expose the ages of all donors, as well as reflect the possible differences between them.

-It is important to describe the type of lipoaspirate and technique used to obtain adipose samples. Because it is known AD-MSC isolated from subcutaneous regions show functional differences than that obtained from the deep layer adipose tissue (Di Taranto et al.,. Cytotherapy 2015, 17, 1076-1089). Likewise, power-assisted liposuction methodologies showed higher proliferative potential and resistance to senescence in isolated AD-MSC than laser-assisted liposuction and surgical biopsy (Bajek et al. Journal of cellular biochemistry 2017, 118, 1097-1107). Likewise, microaspiration of fat with micro-cannulas has been reported to be more efficient than the usual procedures, as expressed in higher yields, greater viability, better adhesion rates, and greater secretion of growth factors, such as insulin-like growth factor (IGF ) and platelet-derived growth factor (PDGF) (Sorrentino et al., Experimental hematology 2008, 36, 1035-1046).

-I think that in the discussion section the authors should extend more in that the effects of the MSCs are mainly due to their paracrine effects, and, thus, highlight the importance of the use of the secrets or their different components, such as exosomes , for future therapies based on MSCs.

Reviewer 2 Report

The study submitted by T Taha et al. takes a bioinformatics approach aiming to describe the cellular response of enriched ASC and liposuctioned white fat tissue (WAT) to TNFA treatment. The authors address a clinically relevant question: whether liposuctioned WAT and ASC behave in a similar fashion under inflammatory conditions (mimicked by exposure to TNFA treatment). The authors identified several genes and biological pathways that undergo cell- or tissue-specific activation by TNFA. However, despite great effort on behalf of the authors to condense and present their data in a respective way, there are several major weaknesses in the study design and presentation which prevent my unprejudiced approval of their work.

Major points:

Study design and conclusion: The design of the study is weak. The authors compared liposuctioned fat tissue harvested from abdominal or thigh subcutaneous tissue treated for 48h in DMEM with gene profiles of ASC cultured on a plastic surface for at least 3 days, enriching a cell type that was not further assessed for stemness or differentiation capacity. So it is not necessarily surprising that a) ASC are different from liposuctioned WAT and b) ASC gene expression is more homogenous than in liposuctioned WAT (figure 3, PC4-analysis). Therefore I urge the authors: To separately mark data from abdominal vs. thigh WAT and ASC (are there differences, probably the outlier in figure 3, PC4 WAT at value 40 are the thigh values?) Discuss the limitations of the study in the discussion section (mixed anatomical origin, ex-vivo expansion vs. direct culture of the tissue) Include information how fat was harvested (system, pressure, cannula diameter, gender, age, BMI of the patients etc. in M&M section) The manuscript is mainly focused on cell numbers differentially regulated in the distinct treatments, but there is hardly any information is edited for the reader to get quick information about relevant regulations. Please include a table describing the most relevant/significant regulations found in deep-sequencing If possible, validate some of the most promising target genes with qPCR to validate sequencing data Cite a paper or proof (in-vitro differentiation or FACS analysis) that the ASC isolation method indeed enriches for ASC Extend section 2.3 by describing the most significant GO-terms/biological pathways identified by the screen. (At present the reader will not take in more information than a) ASC are different from liposuctioned fat tissue and b) TNFA induces a different response between the two. Both are not very surprising conclusions!). Figure 2: Split the “all in one” Venn diagram into more diagrams to ease understanding of the regulations (eg. ASC vs. WAT, ASC-TNFA vs. ASC, etc.) The manuscript is missing functional data that supports the biological importance of the identified regulations. At least for the most relevant regulations the authors will find some publications in the literature that allows more precise conclusion of their data. Introduction is too long - suggest to make it more focus and related to the results part

Minor points:

Line 3: ADSPCs is an unusual term for adipose derived stem cells. Either use “ASC,” or as recommended by iFAT : “ADSC” as uniform term for adipose derived stem cells. Since you are not investigating WAT but liposuctioned WAT, I suggest the use of the term “liposuctioned WAT” instead of “WAT” (because there are significant differences in excised vs. liposuctioned fat tissue!) “ASC 0h” – do samples correspond to ASC expanded for 3 days? Insert this information in the M&M part. Include information on how liposuctioned WAT was processed for 48h culture (washed to get rid of blood components (erythrocytes, etc.) and whether it was also cultured in the presence of 10% FCS) Please insert a table where the most significant TNFA- regulated genes are listed inclusively the extent of regulation. Line 213: the gender of a patient is a “negatively” affecting donor factor - why? English language usage is appropriate and understandable, with only a few minor spelling or grammar errors throughout.

Line 74: “Dependent of their microenvironment…” should read “Dependent on their microenvironment”

Line 99: “is more homogeneously im comparison to WAT” should read “is more homogenous in comparison to WAT”

Line 126: “3129 genes were significantly higher expressed in ADSPCs, when compared to WAT” should read “down-regulated” or “lower” not “higher”

Line 190: “the therapeutical use of isolated stem cells” should read “the therapeutic use of isolated stem cells”

Line 225: “for clinical use because better controllable” should read “because they are better controllable” or “because they behave in a more predictable manner”

Reviewer 3 Report

Stated in methods that participants were healthy individuals but having liposuction, can the authors provide more details? What were sexes and BMIs of participants? High BMI individuals could potentially exist in a chronic state of inflammation, originating from their excess white adipose tissue, which can be modulated by sex.

Which white adipose depots were used in the analysis? Were the thigh and abdominal white adipose depots pooled for each participant or were separate depots used in the analysis? Thigh and abdominal depots are two distinct types, i.e. subcutaneous and visceral which respond differently to inflammatory stimuli. Pre-menopausal age also has an impact on how these depots respond, so again I seek clarification on this. 

What kind of RNA yields were you receiving from tissue and culture?

WAT was plated as complete tissue for 48h rather than disaggregated into it's component cellular structure, how do you know that the tissue didn't undergo any hypoxia or necrosis impacting the results?

Did you run any flow cytometry to confirm the ADSPCs you had extracted were a homogenous population?

Round 2

Reviewer 2 Report

The authors have addressed all the issues raised.

Reviewer 3 Report

Original comments have been answered sufficiently.